# Effects of Heating Rate and Temperature on the Thermal Pyrolysis of Expanded Polystyrene Post-Industrial Waste

**DOI:** 10.3390/polym14224957

**Published:** 2022-11-16

**Authors:** Arantxa M. Gonzalez-Aguilar, Victoria P. Cabrera-Madera, James R. Vera-Rozo, José M. Riesco-Ávila

**Affiliations:** Engineering Division Mechanical, Engineering Department, University of Guanajuato, Campus Irapuato-Salamanca, Guanajuato 37320, Mexico

**Keywords:** thermal pyrolysis, temperature, heating rate, expanded polystyrene waste

## Abstract

The use of plastic as material in various applications has been essential in the evolution of the technology industry and human society since 1950. Therefore, their production and waste generation are high due to population growth. Pyrolysis is an effective recycling method for treating plastic waste because it can recover valuable products for the chemical and petrochemical industry. This work addresses the thermal pyrolysis of expanded polystyrene (EPS) post-industrial waste in a semi-batch reactor. The influence of reaction temperature (350–500 °C) and heating rate (4–40 °C min^−1^) on the liquid conversion yields and physicochemical properties was studied based on a multilevel factorial statistical analysis. In addition, the analysis of the obtaining of mono-aromatics such as styrene, toluene, benzene, ethylbenzene, and α-methyl styrene was performed. Hydrocarbon liquid yields of 76.5–93% were achieved at reaction temperatures between 350 and 450 °C, respectively. Styrene yields reached up to 72% at 450 °C and a heating rate of 25 °C min^−1^. Finally, the potential application of the products obtained is discussed by proposing the minimization of EPS waste via pyrolysis.

## 1. Introduction

Plastics have played a crucial role and have been essential in the evolution of human society for 50 years because they are versatile, light, flexible, and moldable, and their production cost is low. Plastics have promoted the development of numerous applications in the automotive industry, electronics, construction, medicine, and others [1,2].

By 2020, the global production of plastics was estimated at 367 million metric tons [3,4], of which North America (United States, Mexico, and Canada) produces 19%, whereas Mexico individually contributes 2% of the global total [4,5], corresponding to 7.3 million tons approximately, which is annually increasing.

If the production and management of waste are not controlled, approximately 12,000 metric tons of plastic waste will contaminate the environment by the year 2050 [2].

Polystyrene (PS) is heat resistant, lighter in weight, and has good strength and durability, making this polymer suitable for various applications [6]. PS applications include food packaging, beverages, household appliances, the automotive field, and insulating systems for the construction industry [7,8,9,10]. Therefore, the production of PS occupies the fourth place after polyethylene (PE), polypropylene (PP), and polyvinyl chloride (PVC) [11]. The global production of expanded polystyrene (EPS) was 1.7 million tons in 2016, and the world demand for this material is constantly increasing [12]. Recycling and reusing plastic waste are essential for several reasons, the most important being conserving natural resources and reducing environmental pollution. In terms of recycling, Mexico annually consumes 125,000 tons of EPS but only recovers 1% for new-product reuse [13]. In such a scenario, there is great potential for developing an EPS recycling industry.

Plastic recycling technologies began to develop in the 1970s, and since then there have been many advances; plastic recycling can be grouped into four categories: primary, secondary, tertiary, and quaternary [14,15,16]. However, recycling plastics is difficult due to the treatments and processes required to obtain recycled products of acceptable quality. The efficiency of mechanical recycling depends on the residual plastic’s quality and the sorting process’s efficiency (primary and secondary). Materials that cannot be recycled by the mechanical method must be incinerated (quaternary) or landfilled. The high degradation stability and low density of PS cause significant problems when disposed of in landfills; therefore, the processing and recycling of PS waste is a significant problem [17].

Various technologies involving chemical recycling (tertiary) have been researched and developed including depolymerization, pyrolysis, gasification, and hydrocracking [18]. Pyrolysis is a chemical recycling technique that thermally degrades long polymer chains into small molecules in an inert environment or with limited oxygen at high temperatures (300–900 °C) [6]. The pyrolysis process generates three main products: a liquid fraction that can be used as a fuel or that can be processed into value-added chemicals; a gas fraction, which can be used to supplement the energy requirements of the process itself; and finally, a solid carbon by-product that can be used as an energy source due to its relative energy content [6,19,20,21,22]. Pyrolysis is considered an attractive method for recycling plastic waste since it transforms plastics directly into usable energy and valuable products for the chemical industry. On the other hand, the pyrolysis process is friendly to the environment because it does not produce residues [23,24]. However, not all studies have determined the composition or applications of the three resulting products.

The liquid hydrocarbon obtained by pyrolysis is not a standardized product; therefore, there are no official test methods. The liquid hydrocarbon composition varies significantly with the composition of the raw material, making it difficult to develop standards or standardized methods. Even with these challenges, efforts have been made to formalize methods for testing products to achieve a standard outcome.

Styrene is the primary aromatic compound found in the pyrolysis of EPS [17]. Benzene is another compound used to produce cyclohexane and phenol [25]. Toluene is generally used as a gasoline mixture to promote high octane, as a solvent paint, and as a precursor to synthesize other chemical compounds [12]. Additionally, ethylbenzene is widely used to generate styrene and as a solvent [26]. Xylenes and α-methyl styrene are other valuable raw materials for manufacturing plasticizers and resins [27]. In this context, the pyrolysis of EPS is an effective method for obtaining valuable chemical products in the chemical and petrochemical industry [28,29]. Therefore, recovering those chemical compounds from plastic waste is essential to the recycling industry.

As mentioned, the main objective of this study is to evaluate the influence of the reaction temperature and the heating rate on the thermal pyrolysis process of EPS waste in the conversion yields of liquid hydrocarbon, gases, and solid fractions. In addition, the effect of these input variables is compared with the aromatic compounds of the liquid products. Furthermore, the possible application of all the by-products obtained is discussed to propose a process that generates minimal waste. Moreover, the depolymerization mechanism is discussed, and the structure–property–processing relationship is addressed. Therefore, the thermal pyrolysis process of EPS waste is analyzed as a potential for developing a recycling industry in Mexico.

### PS Waste Thermal Pyrolysis Overview

PS is a synthetic aromatic polymer in solid or foam form made from styrene monomer. Standard EPS comprises 98% of air and only 2% of PS [30]. Additionally, EPS waste has a chemically inert behavior, which means that it does not decompose, degrade, or disappear in the environment quickly [31].

Some of the physicochemical properties of PS are listed in Table 1. It is observed that polystyrene is mainly composed of volatile matter with a maximum of 99.59 wt.%. It also has a high carbon content (91 wt.%) and a low hydrogen and oxygen content. PS has a high calorific value; however, its use as a fuel is questionable due to its major aromatic content.

Among the parameters that have the most influence on the pyrolysis process is temperature, since it is the one that controls the cracking reaction of the polymer chain [38,39]. Some studies reported that pyrolysis at low temperatures enhances the formation of the liquid phase and the production of long hydrocarbon chains [38], while others affirm that by increasing the temperature, the yield of liquid hydrocarbon decreases, and the formation of gases increases. High temperatures improve the secondary reactions inside the reactor, which reduce the obtained solid by-product [40].

In terms of state of the art on thermal pyrolysis of PS, the work of Lu et al. [33] is highlighted, where they experimented with virgin PS in a reactor under an inert atmosphere with nitrogen. The process was carried out at a heating rate of 5 °C min^−1^ up to a reaction temperature of 420 °C for 120 min. Their results reported a liquid hydrocarbon yield of 76.26%, and 73% styrene stands out in its composition.

On the other hand, Verma et al. [12] evaluated the influence of temperature (400–700 °C) and heating rate (5–25 °C min^−1^) of thermal pyrolysis of polystyrene waste. The pyrolysis process was carried out in a batch-type reactor and under an inert atmosphere with nitrogen at a flow of 200 mL min^−1^. Regarding the heating rate, the results concluded that the liquid yield increased as the heating rate increased; however, higher rates decreased its yield. The optimum temperature found under these conditions was 650 °C and a heating rate of 15 °C min^−1^, obtaining 94.37% of liquid hydrocarbon. The composition of the liquid hydrocarbon corresponded to a concentration of 84.74% of styrene.

A recent study developed by Van der Westhuizen et al. [35] evaluated the thermal pyrolysis process of three different types of PS to analyze the effect of contamination of the raw material on fuel production. The research studied, as input factors, the temperature and the heating rate in a semi-batch reactor. Additionally, the study industrially scaled the process to a semi-continuous rotary reactor and analyzed the properties of the fuel. The results indicated that contamination in the polystyrene raw material can decrease the liquid yield by up to 6.4%; however, it did not significantly affect its energy content. Van der Westhuizen et al. highlighted the importance of blending PS pyrolytic oil with some other transportation fuel due to its high aromatic content.

## 2. Materials and Methods

### 2.1. Materials

The raw material used in this work was EPS waste, generated during packaging manufacturing. This waste was supplied by a company in Irapuato, Guanajuato, Mexico. Analytical grade reagents (toluene, benzene, ethylbenzene, xylenes, and styrene) were purchased from Sigma-Aldrich.

### 2.2. Thermogravimetric Analysis (TGA)

Thermogravimetric analysis is commonly used to consider the degradation trend in terms of different parameters of the pyrolysis process, such as temperature, heating rate, and others [41]. TGA can be used mainly to study the degradation behavior of polymeric materials, including homopolymers, copolymers, and others [42]. This study performed the thermogravimetric analysis of EPS using a TA Instruments SDT Q600 thermobalance (New Castle, DE, USA). The initial mass of the sample was 4.41 mg. The experiment was carried out under an inert atmosphere with nitrogen gas (N2 5.0) at a 20 mL min^−1^ rate and a heating rate of 20 °C min^−1^ up to 600 °C.

### 2.3. Pyrolytic Oil Characterization

One of the main physical properties of a material is density; in this study, the density was evaluated by buoyancy using a glass hydrometer, and the measurement was performed under the ASTM D 1298 standard, at 20 °C. For this study, a Cannon-Fenske viscometer was used, and the measurement of kinematic viscosity was performed under the ASTM D445 standard at 40 °C. On the other hand, the heating value was determined using an IKA C3000^®®^ isoperibolic bomb calorimeter (Staufen, Germany) under the ASTM D4809 standard.

### 2.4. Gas Chromatography

The qualitative analysis and identification of the chemical compounds present were carried out using a Varian^®®^ 450 GC gas chromatograph (Waltham, MA, USA). The GC was equipped with an Omegawax^®®^ 250 fused silica capillary column, 30 m × 0.25 m × 0.25 μm, using benzene, cumene, styrene, ethylbenzene, and toluene as standards. Helium was used as the carrier gas at a 25 mL min^−1^ flow rate. The injection volume was 1 µL with a split ratio of 1:50. The injection temperature was 250 °C. For temperature programming, the oven was held at a temperature of 40 °C for one minute and then increased to 200 °C at a 10 °C min^−1^ rate; it was then increased to 240 °C at a rate of 5 °C min^−1^ and maintained for 15 min. For the quantitative analysis of the products, C_19_ was used as the internal standard.

### 2.5. Pyrolysis Experimental Setup

This study carried out the thermal pyrolysis of EPS waste in a semi-batch type reactor. The reactor consisted of a stainless-steel tube 17 cm high and 4.5 inches in nominal diameter. The reactor was heated by an external band-type electrical resistance with a ceramic core. Temperatures were monitored using k-type thermocouples inside the reactor and controlled by PID temperature control within ±4 °C.

The experimental scheme of the studied experimental process is shown in Figure 1. The reactor was fed with EPS waste and was hermetically sealed to prevent leaks without the addition of any solvent or any inert gas. The reactor was heated, and the experienced temperatures considered in this study were those obtained through the thermocouple (TC1). The gases produced by thermal pyrolysis were dislodged through a pipe that flowed into a distiller manufactured according to the ASTM D86 standard; this consisted of a pipe submerged in a container with water at room temperature. The gases that were not condensed in the first stage passed through a second countercurrent flow condenser. The condensed product was stored in the secondary collector, the gases that were not condensed in this stage passed to a water trap unit, and, finally, the non-condensable gases were released. The solid fraction was collected from the bottom of the reactor.

Pyrolysis residence time started when the temperature measured by the thermocouple (TC2) reached 150 °C and ended 30 min after the thermocouple (TC1) reached the desired temperature.

### 2.6. Operation Parameters

Statistical analysis was performed with the experimental data obtained in the present study. The analysis was performed using Statgraphics Centurion software XVI with a multilevel factorial design, which is used to study effects with n quantitative factors. The input variables were the heating rate and the reaction temperature; in contrast, the output responses focused on the percentage of conversion to liquid hydrocarbon and the formation of styrene. Both input variables had four levels, and a replica was made for each experiment representing 32 runs.

The yields of liquid, solid, and gas fractions were calculated using Equations (1)–(3), respectively:(1)Liquid yield (wt.%)=(Liquid mass/EPS mass) × 100
(2)Solid yield (wt.%)=(Solid mass/EPS mass) × 100
(3)Gas yield (wt.%)=[(Liquid mass+Solid mass/EPS mass)] × 100

## 3. Results

### 3.1. Thermogravimetric Analysis

Figure 2 shows the calorimetric curve indicating the non-isothermal mass loss of EPS measured by a thermogravimetric analyzer (TGA) at a heating rate of 20 °C min^−1^. The results show that the sample tended to degrade at temperatures higher than 350 °C, obtaining a 98% mass loss at 454 °C, approximately. The final degradation temperature was close to what was reported by Kremer et al. [43], which was 448 °C for PS at the same rate of 20 °C min^−1^. Fuentes et al. [44] reported temperatures close to 458 °C for virgin PS and PS wastes.

As noted, these slight differences between the degradation temperatures of the EPS sample used for the TGA in this work and the ones reported by Kremer et al. [43] and Fuentes et al. [44] were the product of several factors, such as the preparation method of polymer, particle size, the molecular weight of the polymer, operating conditions of the thermogravimetric apparatus, and the mathematical treatment of thermogravimetric data [45,46,47].

Even when those factors influenced the degradation temperatures, activation energy, and kinetic behavior, they did not affect the overall thermal decomposition. The TGA/DTG plots (Figure 2) for the thermal decomposition of the EPS revealed a one-step degradation in the temperature range of 350 °C to 450 °C, which is consistent with the data reported in the literature [46,47,48,49,50,51].

EPS has shown a glass transition temperature of around 100 °C [52], and when thermally decomposed melts at about 160 °C, and the volatility of molten polymer with high molecular weight decreases at 275 °C [47]. This behavior could be seen in the first stage of the exothermic reaction (blue line) when it surpassed 288.20 °C and reached 375.81 °C.

At this point, the single-phase degradation occurred (green line), and it was attributed to the decomposition of the EPS solid matrix to volatile styrene monomers and derivatives (fragments with low molecular weight), reaching 431.66 °C to 454 °C for the complete degradation reported by Mehta et al. [48] and Ali et al. [47]. It is noteworthy to mention that this weight loss indicated that thermal degradation of EPS in a non-inert atmosphere will show a reductive behavior.

### 3.2. Influence of Temperature on the Performance of the EPS Pyrolysis Process

Table 2 shows the conversion yields of the products obtained at different pyrolysis temperatures grouped by the heating rate experimented. It was observed that the production of liquid hydrocarbon increased as the pyrolysis temperature increased. For any heating rate, the maximum liquid hydrocarbon yield was obtained at 450 °C, which was consistent with results from TGA. In this context, with temperatures above 450 °C, the liquid hydrocarbon yield began to decrease by a maximum of 3 wt.%, related to the reduction in the fragments of high molecular weight and the increment of volatile styrene monomers and derivatives (low molecular weight), as well other gas products of the reductive atmosphere.

Additionally, Maafa [6] reported that if the preferred product in the PS pyrolysis process is liquid, it is recommended to use a temperature range of 350 to 500 °C. In contrast, if ash or gases are desired as a product, temperatures above 500 °C are indicated. The yield of gases and solid fractions is significant at a temperature of 350 °C, while there is no significant difference for the other temperatures. This is explained by the fact that in the range of 350 °C and 450 °C, the single-phase thermal degradation of EPS is a radical chain process characterized by three consecutive steps: (1) initiation, (2) propagation, and (3) termination [47,53,54].

During this time, the diffusion of heat or decomposition gases has to be considered as a simultaneous process to the overall chemical reaction (including intermolecular condensation reactions), which has an endothermal/exothermal behavior, inducing heterogeneous temperature distribution in the reactor with no effect on the general composition [55,56].

Particularly in semi-batch systems, like the one used in this study, the evaporated volatiles are removed from the heated zone by evacuation or purge and sweep gases [57] Consequently, thermal degradation occurs only in the liquid phase, and the remnant cracking reactions in the gas phase are negligible [58].

The results of this study were similar to those reported by Tamri et al. [41], who studied the high-impact polystyrene (HIPS) pyrolysis process and obtained maximum yields at 450 °C. In the present study, a maximum yield of 91 wt.% was obtained, having a difference of 3.9%, respective to their study, but without the two additional parameters used by them: an inert environment with nitrogen gas flow and a stirring of 50 rpm. Even compared to catalytic pyrolysis processes, which work at higher temperatures (superior to 600 °C), with the process addressed in this study, it is possible to recover higher values in liquid hydrocarbon [6,59,60,61,62].

### 3.3. Influence of Heating Rate in Obtaining Liquid Hydrocarbon

The heating rate is a parameter that influences the pyrolysis process, directly impacting kinetic behavior; studies have observed that a higher heating rate enhances the production of ashes and gases, reducing the yield of liquid hydrocarbon [63]. Figure 3 shows the results obtained evaluating the four heating ramps (4, 12, 25, and 40 °C min^−1^). It was observed that for the heating rates experienced, both the pyrolysis temperature and the effect of the heating rate had quadratic responses in the yield of liquid hydrocarbon. At a higher heating rate, it will favor the production of liquid; however, it has a maximum point at a 12 °C min^−1^ rate. With higher values, a decrease in performance will be observed. Nanda and Berruti [7] stated that rapid pyrolysis and high degradation temperatures tend to decrease the yield of the plastic pyrolysis liquid. This is due to the faster achieving of the exothermic phase, where the weight loss is at its maximum, which, for the EPS sample used in this work, was 98% at 454 °C, as mentioned before.

Reaching the exothermic phase so fast implies that the endothermic phase, where the polymer suffers the glass transition, has a minimum time to break the larger molecules and melt into a polymer with high molecular weight. Moreover, since the thermal degradation occurs only in this liquid phase and the reactions in the gas phase are negligible [58], the volatile compounds produced are minimal. The last are those that after condensation become a pyrolytic liquid oil, which is the main product of interest in this work.

### 3.4. Influence of Temperature and Heating Rate in Obtaining Value-Added Products

The pyrolysis of EPS is the process where the highest conversion percentage is obtained among all plastics. However, the pyrolytic liquid cannot be used as fuel due to its aromatic composition (principally styrene and α-methyl styrene), which causes a very low thermal–oxidative stability [17] and, therefore, increased engine carbon deposition (if used as automobile fuel) [56]. Thus, the application of this product is mainly based on the petrochemical industry. Figure 4 shows the results of the compounds obtained in the temperatures experimented, grouped in the four heating rate studies. It was observed that for the thermal pyrolysis of EPS in a semi-batch reactor and regardless of the temperature, the main product was a liquid rich in aromatic compounds such as styrene, toluene, ethylbenzene, and α-methyl styrene. The results showed that styrene was the aromatic compound found in the highest proportion. The maximum styrene concentration found was 72.99% at a temperature of 450 °C and a heating rate of 25 °C min^−1^. In contrast, the minimum was found at 51.28% at a temperature of 350 °C and a heating rate of 40 °C min^−1^. This is explained by the fact that in the range of 350 to 450 °C, the single-phase thermal degradation of EPS was occurring, as has been stated before.

Toluene concentrations varied from 4.28 to 12.31% at 350 and 450 °C, respectively. In the chromatographic analysis, values lower than 0.5% were discarded for this study. Additionally, as by-products of the process, the solid fraction was considered to be mainly coal and other non-degradable residues and non-condensable gases. According to the literature, the significant components in the gaseous fraction are related to alkane and alkene. These non-condensable compounds and non-water-soluble gases are methane, ethane, ethene (ethylene), propane, and pentene [63,64,65]. The presence of nitrogen has also been reported [64].

The reaction mechanism of the EPS thermochemical degradation is a chain reaction that begins with a random scission (β-scission) that forms macroradicals (C_13_–C_24_ fraction) along with styrene [66,67,68]. The second stage, called propagation, is a series of intermolecular hydrogen transfers, initially forming low molecular weight macromolecules (C_6_–C_11_ fraction), and in the final stages, dimers and trimers are derived from styrene [55,68,69]. This is the best fitted reaction model, which involves around 2700 to 4500 reactions happening simultaneously, entailing 64 to 93 dead and live species, according to the literature [70,71,72]. Based on the above and the results of the chromatographic analysis of the present study, the proposed reaction mechanism for the degradation of polystyrene is shown in Figure 5.

Toluene, ethylbenzene, and cumene are intermediates between the fractions, donating or accepting hydrogens during the propagation stage because to degrade styrene monomers, dimers, and trimers, many hydrogen radicals are required [73]. This is correlated with the change in styrene selectivity [17], which decreases as the temperature and the heating rate increase [63], as can be seen in Figure 4, where at 500 °C a noticeable fall is shown for the heating rates of 25 and 40 °C min^−1^. Furthermore, as mentioned before, the thermal degradation of EPS only happens in the liquid phase and between the range of 350 to 450 °C, reaching complete degradation at 470 °C [47,48].

Ethylbenzene is a hydrogenated compound derived from styrene, which explains the relationship between both and their mirrored behavior, as shown in Figure 4. As the styrene concentration increased, the concentration of ethylbenzene decreased, and vice versa. In addition, the relationship between α-methyl styrene and cumene had an opposite behavior, because even though the latter is the hydrogenated compound of the former, the presence of cumene was only detected when the concentration of α-methyl styrene increased considerably. This corresponded to the diminution of styrene at the highest temperature (500 °C), where complete degradation of the EPS sample was reached.

On the other hand, the presence of toluene, even when it showed the same tendency as styrene for any temperature, was in a significant percentage at 500 °C, also corresponding with the lowest styrene concentration. This factor had an overall boost of benzene and cumene compounds, which were only shown at this temperature. Consequently, the obtained oil predominantly comprised styrene monomers, toluene, benzene, styrene dimers, and styrene trimers [63,64]. These oxygenated compounds are the product of the reductive atmosphere inside the reactor and the trapped oxygen in the EPS [63].

### 3.5. Characterization of by-Products from EPS Waste Pyrolysis

Table 3 shows the physicochemical properties of the liquid fraction with the highest liquid yield and highest styrene concentration found and is compared with the literature. It was observed that the highest liquid yields resulted at heating rates of 12 to 25 °C min^−1^, and low heating rates (5 °C min^−1^) decreased the conversion to liquid. The results of the present investigation indicated that temperatures above 500 °C decreased the conversion to a liquid fraction. In this context, it was compared with the results of Van der Westhuizen et al. [35], who reported the characterization of the PS pyrolytic oil with the highest yield of 82.5 wt.% at 550 °C, which is 8.5 wt.% lower than that obtained at 500 °C in the present investigation.

On the other hand, no significant difference in styrene formation was observed between temperatures from 420 to 450 °C. Likewise, at higher temperatures, a lower yield percentage was found. However, the objective of the work carried out by Westhuizen et al. was to reduce the styrene concentration to propose it as a fuel, in the case of the physicochemical properties, density, and viscosity decreases at temperatures close to 550 °C. However, the application to be given to the pyrolytic liquid must be considered to indicate the required values. Finally, the energy content of the pyrolytic oil derived from EPS waste is higher than 41 MJ/kg, so its use as an energy source would be advantageous. Nevertheless, when burned, the aromatic content must be considered.

An essential aspect of the pyrolysis of plastics is identifying the applications of all the by-products. Most of the literature focuses on liquid hydrocarbons; however, few have studied the possible applications of non-condensable gases, which are typically burnt [74]. However, converting non-condensable gases into valuable materials can improve the economy of waste-to-liquid production and make thermochemical processing more competitive compared to other recycling technologies.

Singh et al. [63] mentioned and Veksha et al. [75] studied the conversion of non-condensable gases from the pyrolysis of plastics, including PS, into carbon nanotubes (NTCs). NTCs show a diversity of potential applications, including photothermal therapy, photoacoustic imaging, biomedicine, and even an alternative to removing air pollutants [76].

Additionally, Miandad et al. [77] used PS pyrolysis-derived char for the synthesis of nano-absorbent carbon metal double layer oxides (C/MnCuAl-LDOs) through co-precipitation for the adsorption of Congo red (CR) from wastewater. The nano-absorbent from PS pyrolysis was compared with pure carbon (PC), thermally activated carbon (TAAC), and oxidized carbon (Ox-C). Their results reported that C/MnCuAl-LDOs showed maximum adsorption capacity for CR among all the used absorbents.

Moreover, Dogu et al. [78] mentioned that PS, when pyrolyzed with other solid plastic waste, produces a valuable liquid oil and gases that by aromatization or catalytic reforming could produce an aromatic blend with the potential to be used as aviation fuel.

### 3.6. Statistical Analysis

#### 3.6.1. Liquid Hydrocarbon

Figure 6 shows the standardized Pareto chart for the conversion to liquid oil yield with a confidence interval of α = 0.5 and *t* = 2.05 (blue line). It was observed that the factor with the most significant influence on liquid performance was temperature, having a positive effect. As the temperature increased, the liquid yield increased. On the other hand, there was a significant influence on the interaction between temperature and heating rate.

Figure 7 shows the main effects of temperature and heating rate on the conversion to liquid hydrocarbons. The analysis was performed in a second model. The order allowed for a better approximation model for the phenomenon being studied. It was observed that the response of the variables had a curved effect; as the temperature increased, the liquid yield increased. However, temperatures above 470 °C would decrease the conversion of liquid hydrocarbon. On the other hand, the curved effect of the heating rate on the output response was also observed; heating rates greater than 4 °C min^−1^ but less than 12 °C min^−1^ would improve liquid hydrocarbon production.

Additionally, Equation (4) describes the fitted multiple regression model (R^2^ = 80.40%) of the conversion to liquid hydrocarbon, where *T* is the reaction temperature, and *H* is the heating rate.
(4)$$\%Liquid=-67.3048+0.649545\times T+0.805674\times H-0.0006625\times T^{2}\;-0.00134173\times T\times H-0.0066103\times H^{2}$$

#### 3.6.2. Styrene Production

This section shows the statistical analysis results of styrene production as an output response, evaluating the reaction temperature and the heating rate as input variables. Figure 8 shows the standardized Pareto chart for styrene production by an analysis with a confidence interval of α = 0.5 and *t* = 2.05 (blue line). It was observed that the factor with the most significant influence on the formation of styrene was the heating rate. In addition, it was visualized that this effect was negative, representing that at a higher heating rate, less styrene will be formed. On the other hand, the reaction temperature also influenced the output response; however, its effect was positive. At a higher reaction temperature, the liquid hydrocarbon composition would have a higher concentration of styrene. There was no interaction between both factors that influenced the formation of styrene.

In the same way as the conversion to liquid hydrocarbon, the main effects of the input variables on the production of styrene were analyzed. It is visualized in Figure 9 that both temperature and heating rate had curved effects on the output responses. It is crucial to analyze nonlinear models since sometimes the linear trend makes false statements. Therefore, it was statistically stated that there was a decrease in styrene production at temperatures greater than 450 °C and at heating rates above 17 °C min^−1^.

Regarding the multiple regression model that describes the formation of styrene, Equation (5) describes the behavior with a fit of 77%, where *T* is the reaction temperature, and *H* is the heating rate.
(5)$$\%Styrene=-91.8019+0.65818\times T+1.43878\times H-0.000694625\times T^{2}\;-0.0013590\times T\times H-0.0239486\times H^{2}$$

Finally, in Figure 10, the response surface of the styrene production is shown through a contour plot. It is seen that the highest concentration of styrene in liquid hydrocarbons obtained by thermal pyrolysis of EPS was centered at temperatures below 500 °C and maximum heating rates of 20 °C min^−1^.

## 4. Conclusions

The influence of temperature and heating rate in the thermal pyrolysis process of (EPS) post-industrial waste shows an intrinsic relationship between the production of liquid hydrocarbon yield and styrene content. Moreover, neither the temperature nor the heating rate showed an influence on the general composition found in the pyrolytic oil.

The results showed a maximum yield of liquid hydrocarbon of 91 wt.%, with 72.99% of styrene in its composition, at 450 °C and a heating rate of 25 °C min^−1^. It is noteworthy that in a simple thermal pyrolysis process as the one addressed in this study, it was possible to recover higher values in liquid hydrocarbon compared to other studies. It was even compared to catalytic pyrolysis processes, which achieve these yields at higher temperatures (superior to 600 °C) and by adding elements that lead to higher operational costs [6,59,60,61,62].

This proposed pyrolysis can go hand-in-hand with EPS mechanical recycling, which will solve the limitations of this type of plastic waste management but also for producing a liquid hydrocarbon with added value for the petrochemical industry.

## Figures and Tables

**Figure 1 polymers-14-04957-f001:**
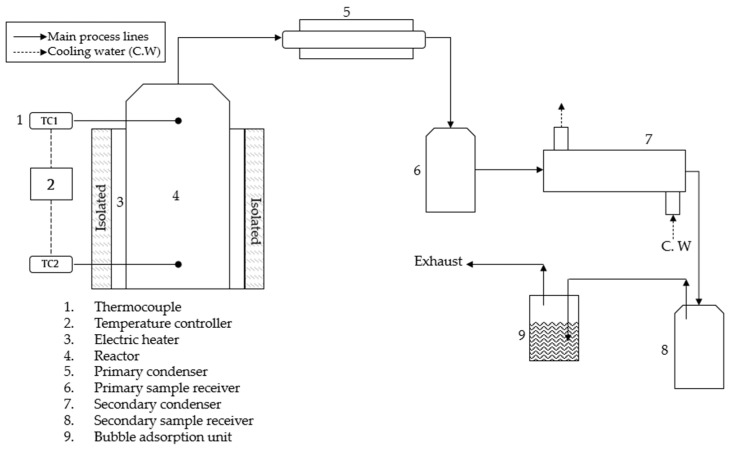
EPS pyrolysis experimental scheme.

**Figure 2 polymers-14-04957-f002:**
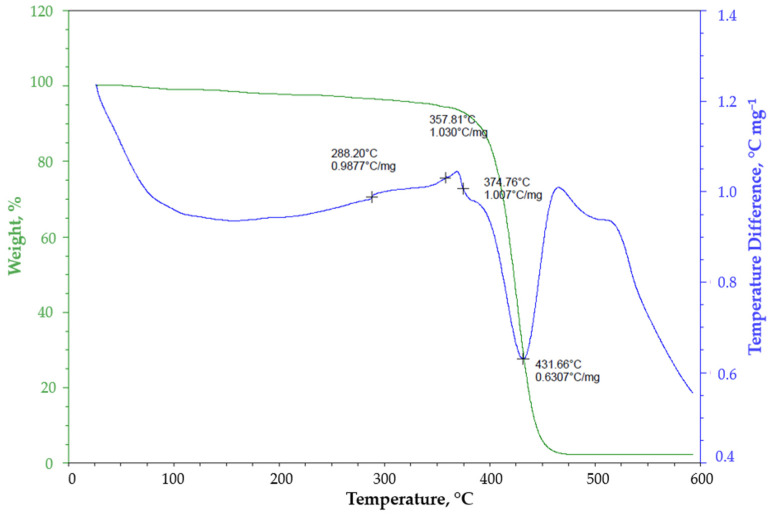
EPS TGA curve at a heating rate of 20 °C min^−1^. Measuring points (+).

**Figure 3 polymers-14-04957-f003:**
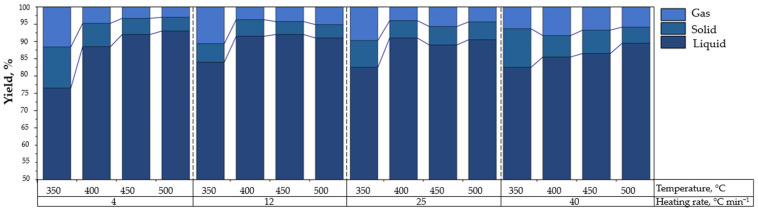
Influence of heating rate on EPS performance.

**Figure 4 polymers-14-04957-f004:**
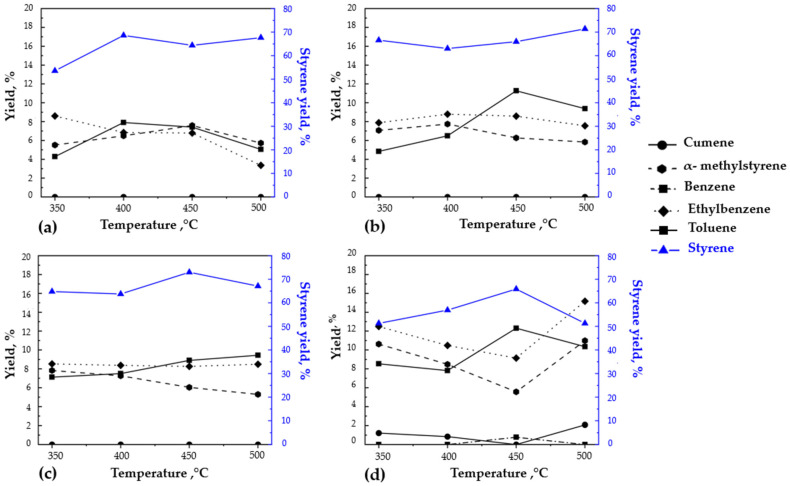
Influence of temperature and heating rate on the obtaining of value-added products. Heating rates: (**a**) 4 °C min^−1^, (**b**) 12 °C min^−1^, (**c**) 25 °C min^−1^, (**d**) 40 °C min^−1^.

**Figure 5 polymers-14-04957-f005:**
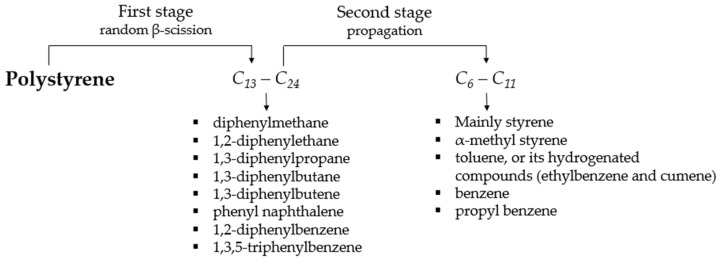
Reaction mechanism for the degradation of polystyrene.

**Figure 6 polymers-14-04957-f006:**
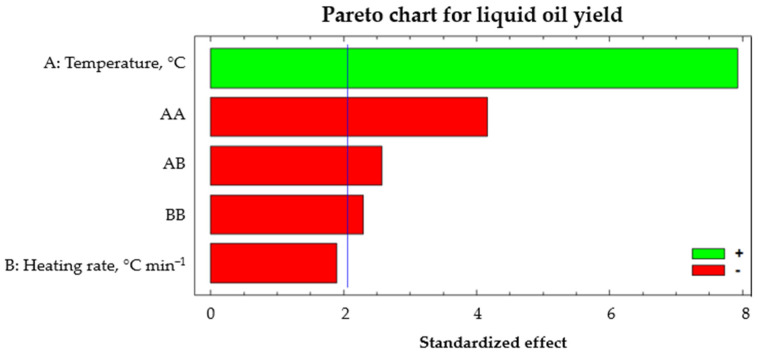
Standardized Pareto diagram for liquid oil yield.

**Figure 7 polymers-14-04957-f007:**
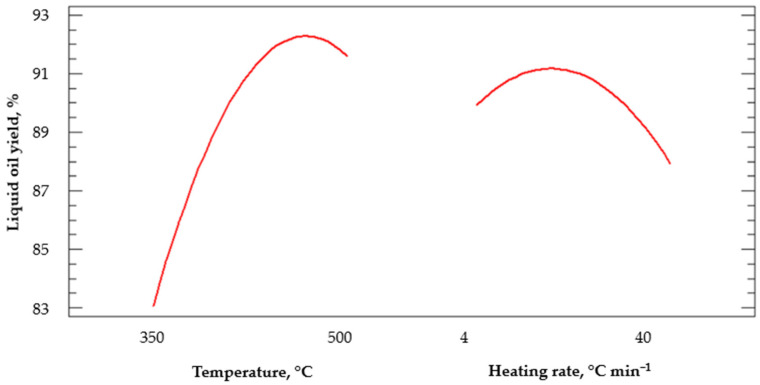
Main effects on the liquid oil yield.

**Figure 8 polymers-14-04957-f008:**
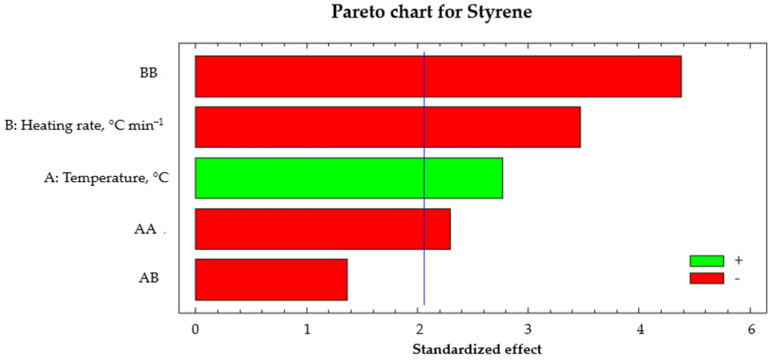
Standardized Pareto chart for styrene production.

**Figure 9 polymers-14-04957-f009:**
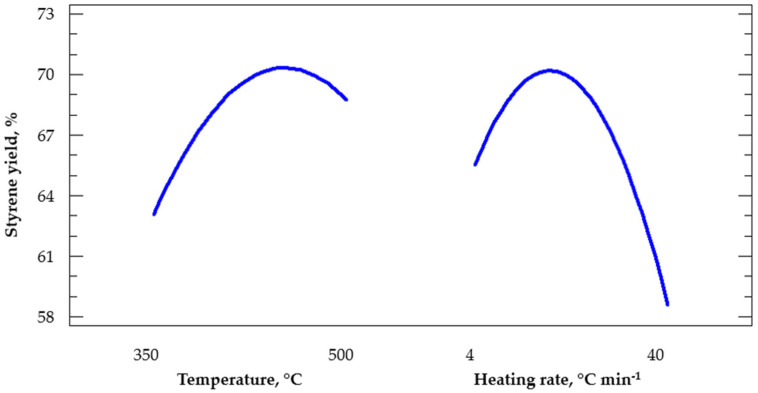
Main effects on styrene production.

**Figure 10 polymers-14-04957-f010:**
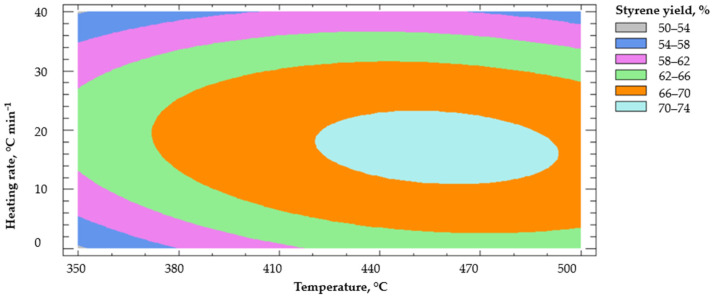
Response surface.

**Table 1 polymers-14-04957-t001:** Summary of physicochemical properties of PS raw material.

Parameter	Value	References
Density, kg/m^3^	1040–1050	[17,32]
Melting point, °C	180–260	[17,33]
HHV, MJ/kg	37.22–42.1	[34,35]
Ultimate analysis, wt.%		
C	66.47–92.7	[25,32,34,35,36,37]
H	7.4–9.43	[25,32,34,35,36,37]
N	0–0.8	[25,34,35,36,37]
S	0–0.51	[25,34,35,36,37]
O	0–6.8	[25,34,35,36,37]
Proximate analysis, wt.%		
Moisture	0–0.24	[25,34,35,36,37]
Volatile	88.9–99.59	[25,32,34,35,36,37]
Ash	0–4.6	[25,34,35,36,37]
Fixed carbon	0.1–2.25	[25,32,34,35,36,37]

**Table 2 polymers-14-04957-t002:** Influence of temperature on the performance of the EPS pyrolysis process.

**Temperature °C**	**Yield, wt% ^1^**					
	**Heating Rate: 4 °C min^−1^**			**Heating Rate: 12 °C min^−1^**		
	Liquid	Gas	Solid	Liquid	Gas	Solid
350	76.5	11.6	11.9	84	10.64	5.36
400	88.5	4.75	6.8	91.5	3.69	4.81
450	93	3.01	4	92	4.24	3.76
500	92	3.3	4.7	91	5.12	3.88
**Temperature °C**	**Heating rate: 25 °C min^−1^**			**Heating rate: 40 °C min^−1^**		
	Liquid	Gas	Solid	Liquid	Gas	Solid
350	82.5	9.71	7.8	82.5	6.34	11.2
400	90.5	4.35	5.2	85.5	8.29	6.2
450	91	3.98	5	89.5	5.84	4.7
500	89	5.66	5.3	86.5	6.74	6.8

^1^ Yield variation of ±1%.

**Table 3 polymers-14-04957-t003:** Physicochemical properties of liquid hydrocarbon from EPS pyrolysis.

	EPS Pyrolysis-Derived Oil		Lu et al. [33]	Van der Westhuizen et al. [35]
**Parameters**				
Temperature, °C	450	500	420	550
Heating rate, °C min^−1^	25	12	5	n.r.
**Yields, wt.%**				
Liquid hydrocarbon	89	91	76.24	82.5
Gas	5.66	3.88	10.75	3
Solid	5.35	5.12	13.01	0.4
Styrene	72.99	71.38	73	39.4
**Properties**				
Density at 15 °C, kg m^−3^	933	935	n.r.	923
Kinematic viscosity, mm^2^ s^−1^	1.09	1.17	n.r.	0.88
Heating value, MJ kg^−1^	41.64	41.65	n.r.	n.r.

n.r. = not reported.

## Data Availability

Not applicable.

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
