# Peer review of "Effects of Heating Rate and Temperature on the Thermal Pyrolysis of Expanded Polystyrene Post-Industrial Waste"

_polymers, 2022, doi:10.3390/polym14224957_

Round 1

Reviewer 1 Report

1.       Introduction part: Please reorganize it, too many paragraphs.

2.       Conclusions: There are too many paragraphs in the conclusions part of this work, please reorganize it.

3.        Figure 4. Please label the subplot as a, b, c, d, or another format.  What is the reason that there are four components in a, b, c, but six components in d?

4.       Figure 3. How to explain that the solid ratios are ranked as heating rate 4C> 40C> 25C> 12C at temperature 350C? while it is not at other temperatures.

5.       What is the innovation point of this work?  

Reviewer 2 Report

This paper introduced “Effects of Heating Rate and Temperature on the Thermal Pyrolysis of Expanded Polystyrene Post-Industrial Waste”. The author has done a lot of work. I think this paper is interesting, but there are many weaknesses, so it is better to revise it carefully.

Several weaknesses of the paper:
1. Introduction:

(1)    The author introduces detailed work and should also highlight the innovative and meaningful nature of the article.

(2)    Line 61: “Plastic waste recycling technology could be grouped into four categories: mechanical recycling, chemical recycling, incineration, and pyrolysis”. From the introduction, it appears that pyrolysis is one of the chemical recycling methods, so this sentence may not be appropriate.

(3)    Line 103: “the pyrolysis process is friendly to the environment because it does not produce residues”. Production residues from pyrolysis were present in most experiments, so this conclusion may need to be revised.

2. Table 1: The density value is too large, please check the parameters. In addition, the molecular formula of EPS is (C8H8)n, and ultimate analysis shows that C, H, N, S and O may be present, please explain the reason.

3. Section 3.1: Heating rate affects pyrolysis behavior, why do authors select the heating rate of 20 ℃/min? why is only one heating rate chosen?

4. Lines 249 and 250: “such as the preparation method of polymer, particle size, the molecular weight of the polymer, operating conditions, and mathematical treatment of thermogravimetric data”. Please explain how these factors affect pyrolysis.
